# Fifteen Years of Continuous High-Resolution Borehole Strainmeter Measurements in Eastern Taiwan: An Overview and Perspectives

Alexandre Canitano [1,*,†], Maxime Mouyen [2,†], Ya-Ju Hsu [1,†], Alan Linde [3,†], Selwyn Sacks [3,†] and Hsin-Ming Lee [1,†]

1 Institute of Earth Sciences, Academia Sinica, Taipei 115024, Taiwan; yaru@earth.sinica.edu.tw (Y.-J.H.); leehm@earth.sinica.edu.tw (H.-M.L.)
2 Department of Space, Earth and Environment, Chalmers University of Technology Onsala Space Observatory, SE-439 92 Onsala, Sweden; maxime.mouyen@chalmers.se
3 Department Terrestrial Magnetism, Carnegie Institution of Washington, Washington, DC 20005, USA; alinde@dtm.ciw.edu (A.L.); sacks@dtm.ciw.edu (S.S.)
* Correspondence: canitano@earth.sinica.edu.tw
† These authors contributed equally to this work.

**Abstract:** As one of the most sensitive instruments for deformation monitoring in geophysics, borehole strainmeter has the capability to record a large spectrum of tectonic and environmental signals. Sensors are usually deployed near active faults and volcanoes and provide high-resolution continuous recordings of seismic and aseismic signals, hydrological variations (rainfall, groundwater level) and natural hazards (tropical cyclones, landslides, tsunamis). On the occasion of the 50th anniversary of the installation of the first *Sacks–Evertson* borehole strainmeter, in central Japan, we present an overview of the major scientific contributions and advances enabled by borehole strainmeter measurements in Taiwan since their installation in the mid 2000s. We also propose a set of future research directions that address recent challenges in seismology, hydrology and crustal strain modeling.

**Keywords:** borehole strainmeter; crustal deformation; tidal calibration; seismic source modeling; tropical typhoon; aseismic deformation; infrasound; stress modeling; high-resolution monitoring; Taiwan

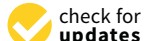



## 1. Introduction

The measurement of strain change in the earth is fundamental to the understanding of long-term tectonic deformation, seismic and aseismic processes, tidal deformation or atmospheric and hydrological perturbations. Developed in the late 1960s to early 1970s to complement existing geodetic techniques (e.g., long baseline, tilt or leveling measurements), borehole strainmeters represent a key component for continuous monitoring of crustal deformation in seismically and volcanically active regions. Installed in deep boreholes in order to isolate them from surface noise sources, borehole strainmeters offer an unprecedented resolution better than 1 ppb (part per billion or nanostrain ($n\epsilon$)), which outclasses detection sensitivity of other geodetic techniques by one order of magnitude or more for periods of seconds to months [1] (Figure 1).

Located at the junction between the Eurasian Plate and the Philippine Sea Plate, the island of Taiwan is a very active seismic and tectonic region. A large fraction (>50%) of the 8–9 cm·yr$^{-1}$ [2] of the Philippine Sea Plate oblique convergence toward the Eurasian Plate in the direction N306° is accounted by the Longitudinal Valley (LV) [3]. Running along the eastern side of the LV, the Longitudinal Valley fault (LVF) represents the major active structure in eastern Taiwan. The fault displays a complex slip regime in which elastic strain is released via a whole continuum of slip modes ranging from destructive earthquakes

to shallow creep [4] and episodic slow slip events (SSEs) [5,6]. The LV is also frequently impacted by tropical typhoons, heavy rainfall and landslides, and thus offers a remarkable natural laboratory to study tectonic, hydrological, subsurface and oceanic deformations and their short- to long-term interactions.

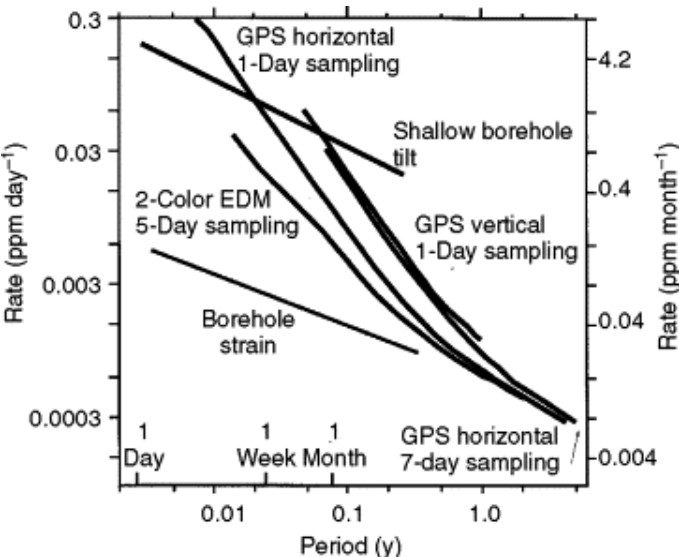

**Figure 1.** Comparative resolutions of crustal strain rates for borehole strainmeters (in part per million (ppm)), differential GNSS, EDM (Electronic distance measurement), shallow borehole tiltmeters as a function of periods [1]. Note that GNSS and EDM assume an 8 km baseline. Borehole strainmeter sensitivity exceeds this of other instruments by at least an order of magnitude for periods of seconds to months.

Following their first installation in central Japan [7] in 1971, *Sacks–Evertson* borehole strainmeters have been widely deployed in seismically active regions, in particular in Iceland [8], the western USA [9] in central Greece [10,11]. They are also a main component for volcano monitoring systems in Italy and allowed detection of volcano deformation [12,13], pre-eruptive signals [14,15] and eruption dynamics [16,17]. Beginning in 2004, to shed additional light on the nature of the deformation in eastern Taiwan, 11 *Sacks–Evertson* [7] borehole strainmeters were deployed as a complement to GNSS stations through a collaboration between the Institute of Earth Sciences Academia Sinica and the Department of Terrestrial Magnetism (now Earth and Planets Laboratory), Carnegie Institution of Washington (Figure 2). During the last 15 years, borehole strainmeters have recorded a large spectrum of tectonic [18,19] and environmental [20–22] signals, allowing us to decipher the mechanisms of processes previously unrecognized in the region. We present here a broad, detailed overview of the main advances enabled by continuous high-resolution strain measurements in eastern Taiwan.

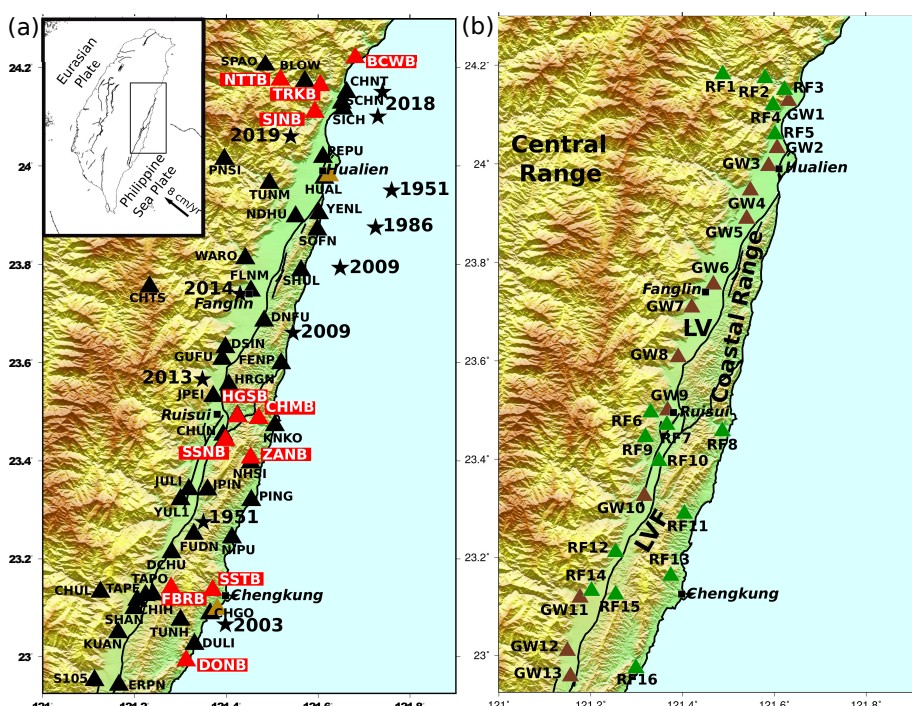

**Figure 2.** Map of station locations, topography, major faults and major cities in eastern Taiwan. (**a**) Earth and ocean deformation monitoring: red triangles indicate *Sacks–Evertson* borehole strainmeter (BCWB, NTTB and CHMB are permanently shut off in 2021), black triangles denote continuous GNSS stations and brown triangles indicate tide-gauges operated by the Central Weather Bureau. Black stars denote major earthquakes in the region since 1950, from north to south: 2018 Hualien $M_w$ 6.1 and $M_w$ 6.4 sequence, 2019 $M_w$ 6.1 Hualien event, 1951 $M_L$ 7.3 Hualien-Taitung mainshock, 1986 $M_s$ 7.8 event, 2009 $M_w$ 6.4 Hualien event, 2014 $M_L$ 5.9 Fanglien event, 2009 $M_w$ 6.1 event, 2013 $M_w$ 6.2 Ruisui event, 1951 $M_L$ 7.3 Hualien-Taitung largest aftershocks, and 2003 $M_w$ 6.8 Chengkung earthquake, respectively. (**b**) Monitoring of hydrological perturbations: brown triangles are groundwater level stations deployed by the Central Weather Bureau and the Water Resources Bureau in Taiwan and green triangles denote rainfall stations operated by the Central Weather Bureau.

## 2. Instrumentation Design and Basic Processing

### 2.1. Sacks–Evertson Sensor: Details and Installation

A *Sacks–Evertson* strainmeter is a hydraulic sensor built upon Benioff's work [23] that suggests that dilatational strain might be measured by recording liquid flow within a buried large container with a small opening. The instrument sensing unit is a cylindrical tube of about 3 m long log by 7 cm diameter filled with silicon oil and connected to a small diameter bellows. The cylinder deformation forces the oil to flow in or out of the attached bellows and the displacement of the top section of the bellows is monitored by a differential transformer (DT1). For a given strain, the bellows moves a distance proportional to the sensing volume divided by the bellows cross-sectional area. To keep the strainmeter within its operating range over indefinite time intervals, a valve is opened for a few seconds as needed to allow oil to flow to or from a reservoir that is decoupled from the strain field [7] (Figure 3). A second, larger sensing unit is present to collect oil flow when valve 1 is opened. Valve 2 allows oil to flow between the second bellows (DT2) and the unstressed reservoir. The control system does not allow both valves to be open at the same time, so that the sensing volume is always closed from the reservoir, thus allowing an uninterrupted record of strain variations. The sensor is deployed in competent rocks (ideally away from any gross cracks, active aquifers, or high topography) and grouted into the borehole using an expansive grout to ensure it is always under compression (Figure 4).

In the LV, sensors are deployed at depths ranging from 170 to 270 m mostly in breccia and marble rocks in three networks near Hualien, Ruisui, and Chengkung (Figure 2). Most of stations are installed at the sea-level altitude, a few kilometers away from the Pacific coast or near rivers (e.g., TRKB, NTTB, CHMB). Stations BCWB, NTTB and CHMB are no longer operational in 2021. Each network is composed of borehole dilatometers monitoring rock volume change (dilatation $\epsilon_v$) and one 3-component strainmeter (TRKB, SSNB and DONB) which allows to resolve horizontal shear strain (differential extension ($\gamma_1$) and engineering shear ($\gamma_2$)) in addition to dilatation. The strain components resolved by the sensors can be stated as follows:

$$\epsilon_v = \epsilon_{ee} + \epsilon_{nn} + \epsilon_{zz}$$
$$\gamma_1 = \epsilon_{ee} - \epsilon_{nn} \tag{1}$$
$$\gamma_2 = 2\epsilon_{en}$$

where $\epsilon_{ee}$, $\epsilon_{nn}$ and $\epsilon_{zz}$ are the east, north and vertical strain components, respectively, and $\epsilon_{en}$ is the east-north shear strain component. To continuously record atmospheric pressure variations, stations are equipped with *SETRA 278* absolute microbarometers. They are sensitive to pressure variation from DC to 2 Hz and measure absolute pressure with a resolution larger than 0.5 Pa, a long-term stability of 10 Pa·yr$^{-1}$, and a dynamic response <0.1 s.

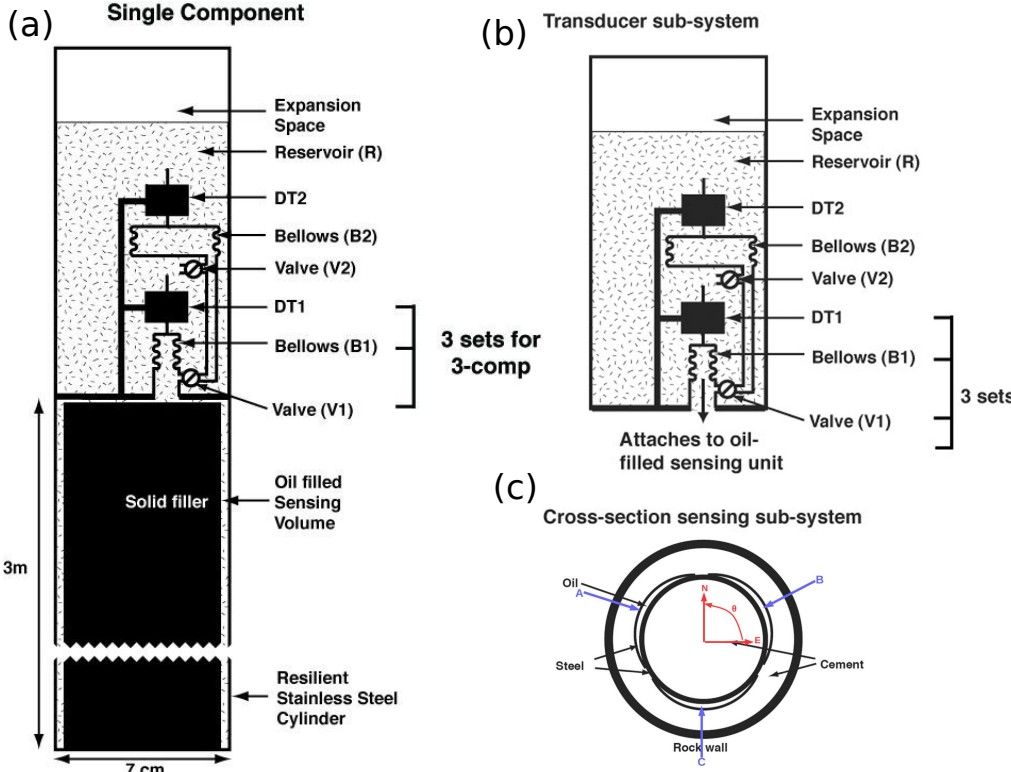

**Figure 3.** Design of a *Sacks–Evertson* borehole strainmeter: (**a**) dilatometer (single-component) and (**b**) 3-component strainmeter [21]. The 3-component design uses the same principle employed in the single-component instrument but incorporates three separate sensing volumes. Note that when valves V1 and V2 are closed, DT2 monitors volumetric changes of a fixed mass of oil that is decoupled from the strain field which thus represents a measure of the temperature at the installation depth (with resolution < $10^{-4}$ °C). (**c**) Cross-section of a 3-component strainmeter. Gauges B and C are 120° clockwise and 120° counterclockwise from gauge A, respectively. The N-E axis reference for strain calculations is in red.

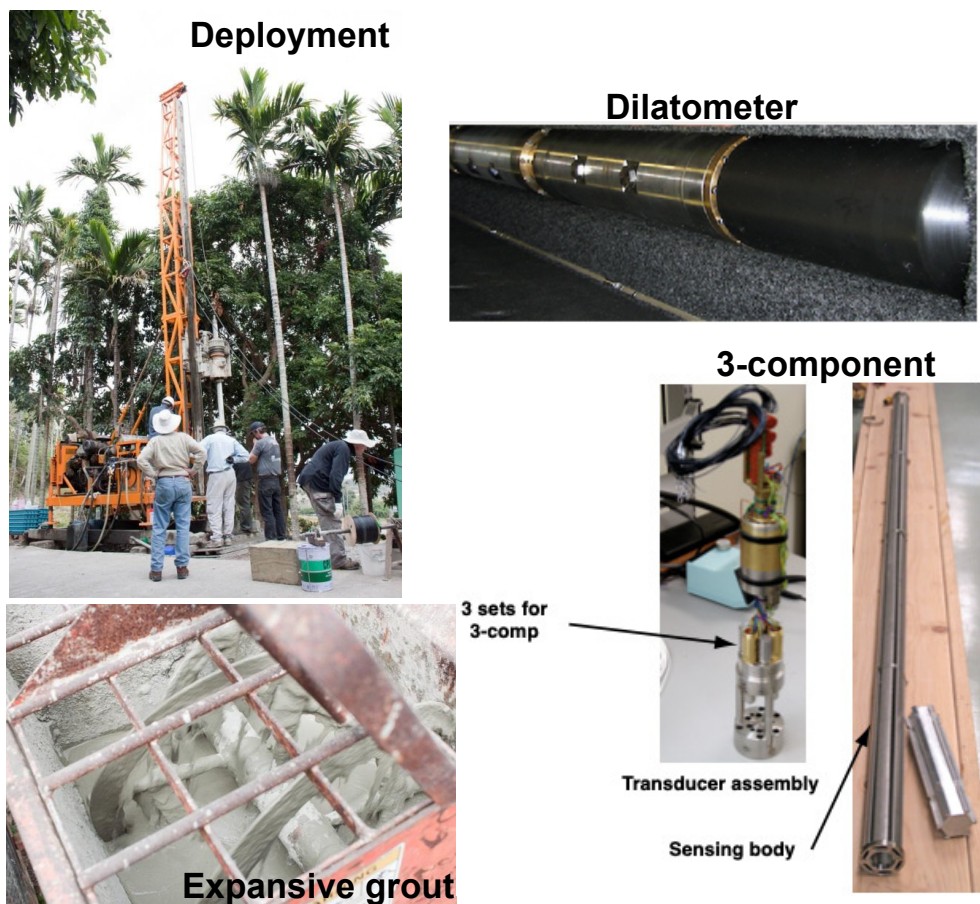

**Figure 4.** Photos of sensor designs and their deployment in the LV.

During the following years after its installation in the borehole, the sensor is affected by transient effects due to curing of the expansive grout and viscoelastic stress relaxation around the newly drilled borehole [24]. They induce exponential trends on the strain records, yet relatively easy to predict and correct (Figure 5).

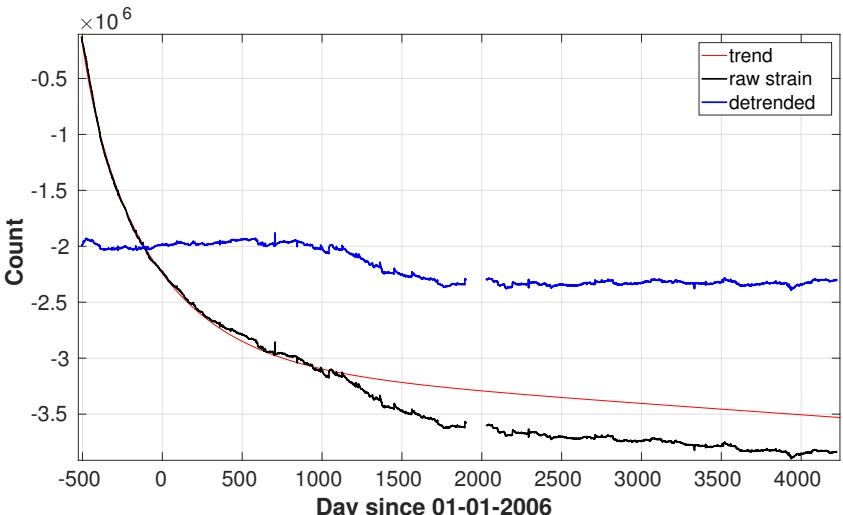

**Figure 5.** Example of exponential trend induced by grout curing and stress relaxation recorded by station ZANB. Expansion is positive.

## 2.2. Sensor Calibration and Orientation

Since the elastic moduli are different between the instrument, grout and host medium, in-situ calibration is a fundamental step to relate the sensor measurements to the regional (far-field) strain. The most common technique is tidal calibration which is based on the recorded amplitudes and phases of $M_2$ (12.42 h period) and $O_1$ (25.82 h period) tidal constituents against synthetic tides [25]. $M_2$ and $O_1$ tides are generally selected because they have the largest amplitudes and relatively little contamination from thermal signals. While dilatational strain is usually accurately recorded by the sensors, deriving robust horizontal shear strain signals from 3-component strainmeters is not straightforward [9]. Shear signals are obtained by differentiating strain gauges and are highly dependent on the orientation of the gauges (poorly resolved during installation), which makes them less robust than dilatation, that depends on the sum of the three strain gauges, irrespective of their orientation. Shear signals are also more vulnerable to cross-coupling due to internal inhomogeneity and topography [25].

Traditional calibration methods present an observation equation for tides equals a linear combination of the reference solid and ocean tides [26] for each tensor strain component ($\epsilon_v$, $\gamma_1$, $\gamma_2$) and then estimate the weight of each component to fit the observations [25,27,28]. To calibrate and estimate the orientation of 3-component sensors in Taiwan, we proceed 'backwards'. We express the reference tides for each component as a linear combination of the three observed gauge data (Figure 3c), we allow different coupling coefficients between gauges to produce a substitute for cross-coupling and estimate the relative contribution of each gauge. In other words, we reconstruct the theoretical shear signals through linear regressions on strain gauge signals, with variable sensor azimuth [29]. The protocol relies on a large range of tidal constituents ($M_2$, $S_2$, $N_2$, $K_2$, $O_1$, $Q_1$, $K_1$, $P_1$) for a robust modeling, and reliable calibrations are inferred for SSNB and TRKB sensors (DONB remains to be calibrated) (Figure 6). Dilatation component is calibrated by correlating the observed tidal waveforms to the synthetic tides and a good agreement with standard method [25] is obtained. Overall, tidal calibration is limited by the resolution of the theoretical reference tides which can yield large uncertainties (up to 30%) [30], in particular for sites deployed in coastal regions. Therefore, other techniques, such as using seismic signals [31], should be tested in the future. Finally, since the tidal response for each site is well understood, the strain time-series are corrected for solid and ocean tides by adjusting a phase and amplitude to the various tidal constituents of the signal [32].

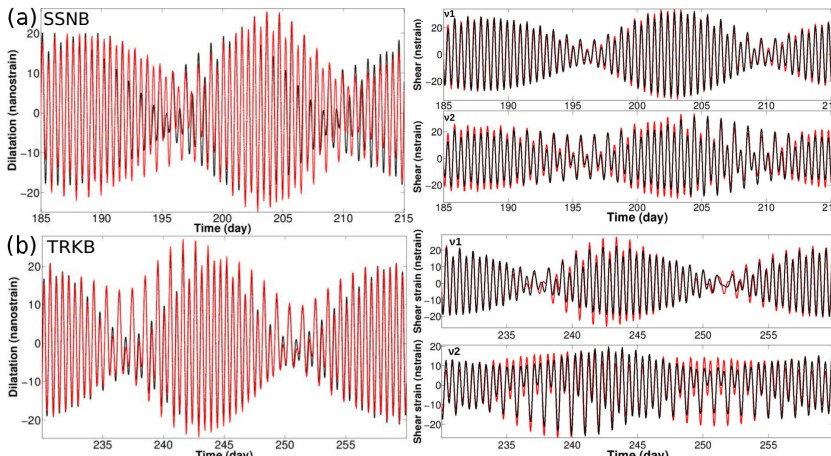

**Figure 6.** Calibration of dilatation (left) and shear signals ($\gamma_1$, $\gamma_2$) (right) for 3-component stations SSNB (**a**) and TRKB (**b**). Black curve denotes theoretical Earth and ocean tides and red curve denotes modeled tides, respectively, over a one-month period. Reprinted from [29] with permission from Springer Nature.

### 2.3. Strainmeter Response to Atmospheric Pressure Variations

Atmospheric pressure variations induce deformation in the Earth's crust that is continuously recorded by borehole strainmeters. The sensor response is instantaneous, that is mainly frequency independent, but strongly site dependent. A decrease of barometric pressure results in expansive dilatational strain and vice versa. Assuming a linear relationship (elastic effect) between barometric pressure and strain variations, barometric pressure admittances for dilatation and shear strain are ranging from about $-0.5$ to $-4$ n$\epsilon$·hPa$^{-1}$ among the stations. Estimates are slightly larger than theoretical values ($-0.7$ to $-1.1$ n$\epsilon$·hPa$^{-1}$) [21], possibly due to errors in calibration and to the use of a simple elastic medium (Poisson's ratio $\nu = 0.25$, rigidity $G = 30$ GPa) for calculations. Since air pressure effects modulate the strain variations, inducing substantial deformation for some sites, accurately estimating and correcting air-pressure-induced strain (Figure 7) is fundamental to detect previously hidden tectonic or non-tectonic signals. Nonetheless, the continuous measurements of crustal strain and atmospheric variations by collocated stations also provide critical insights into the coupling between the solid Earth and atmosphere during tectonic and environmental perturbations. For instance, we present thereafter observations and modeling of the subsurface deformation induced by tropical typhoons and also the first analyze of strain-atmosphere coupling during infrasonic events.

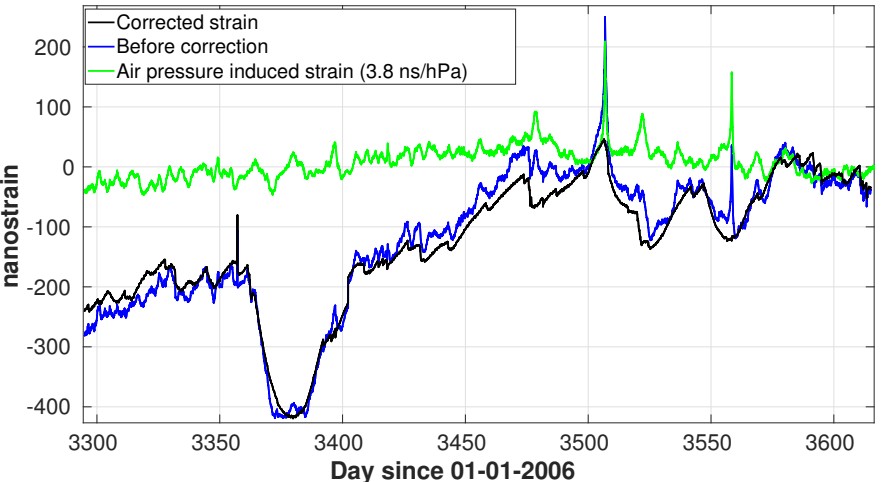

**Figure 7.** Example of dilatation (detrended, detided and calibrated) recorded by SSTB before and after processing of air pressure-induced strain. Expansion is positive. Note that atmospheric pressure signal is also detided.

## 3. Borehole Strainmeter Observations in Eastern Taiwan

Through fifteen years of continuous monitoring of crustal deformation in the LV, a large variety of tectonic and non-tectonic signals were recorded by borehole strainmeters, yielding to a better understanding of processes previously unrecognized in the region. The large spectrum of detected signals illustrates the high-sensitivity of continuous strain measurements that spans time-periods ranging from seconds (seismic processes) to decades (long-term tectonic and environmental processes).

### 3.1. Earthquake Source Modeling: The 2013 Ruisui Earthquake

Strainmeters have the capability to record simultaneously dynamic and static elastic deformations resulting from an earthquake. Dynamic strain waves, that depend mainly on the rupture process and on the crustal structure, are faithfully recorded by the sensors [7]. Static strain offsets reflect the abrupt change in stored elastic energy associated with earthquake elastic rebound. They show little sensitivity to the earth structure and are mainly dependent on the source position and source fault parameters. They are also the most difficult signals to resolve with strainmeters because the rock around the sensor has to respond elastically during high accelerations due to the passage of seismic waves.

Despite their potential for monitoring and constraining earthquake rupture, near-source strain signals have long been overlooked [33]. Figure 8 depicts the dilatation signals recorded by strainmeters deployed in the near-source region of the 2013 Ruisui earthquake (Figure 2). Due to its proximity with operating stations (20–30 km), this earthquake represents one of the best example of volumetric strain detection during a moderate to large event [18] to date. Dynamic strain signals with trough-to-peak amplitudes exceeding 10 $\mu\epsilon$ and substantial permanent coseismic static offset signals (down to $\sim -1$ $\mu\epsilon$) are recorded during the mainshock. Further, the large near- and intermediate strain field (contractional strain $<-1$ to $-3$ $\mu\epsilon$ for stations HGSB and SSNB) that initiates between *P*- and *S*-wave arrivals is well resolved by stations. Canitano et al. (2017) [34] modeled earthquake near-fault displacements using the discrete wavenumber representation method and a simple rupture model derived from seismology and geodesy [18]. They estimated synthetic volumetric strain by finite-difference estimates of displacement time-series. Despite a quite crude source fault parametrization, justified by the small amount of stations, simulations explained reasonably well the largest dynamic strain (at the period of 4–5 s) and the permanent near- and intermediate-field static offsets (discrepancies with observations are $<20\%$).

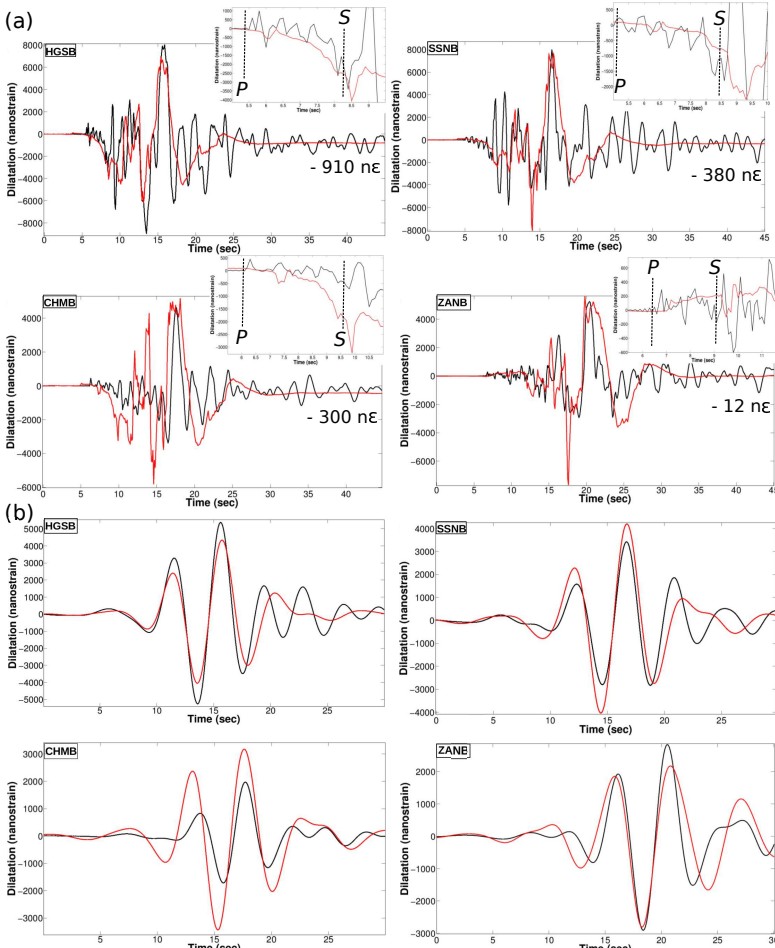

**Figure 8.** (**a**) High-sampling rate (100 Hz) dilatation signals recorded by stations located in the Ruisui network during the 2013 $M_w$ 6.2 Ruisui earthquake (black curve) and simulated time-series using the discrete wavenumber representation method and a simple rupture model (5 point sources) (red curve). Top inset is a zoom on the near- and intermediate field initiating between *P* and *S* wave arrivals (vertical dashed lines). Expansion is positive. (**b**) Recorded dilatation signals (black curve) and modeled signals (red curve) filtered between 3 to 7 s. Modified from [34] with permission from Springer Nature.

In general, detection of zero-frequency near-field signals is a rare occurrence during moderate to large events. Direct detection of near-field signals may allow the measurement of the permanent deformation resulting from faulting [35] as near-field terms are sensitive to fault mechanism and depth but not to the crustal structure. However, since the seismic source radiates continuous direct *P*- and *S*-waves, their large amplitudes, even at low frequencies, make near-field observations complicated. Besides, small to moderate shocks ($M_w \sim 4$ to 6) generate surface displacements below GNSS detection theresold (a few mm). Near-source displacements < 0.1 mm are predicted during the 2013 Ruisui event [34], yet well detected by borehole strainmeters. Strainmeters are also useful to detect possible pre-earthquake strain anomalies [36,37]. Seconds before the Ruisui event, strain is stable at the $10^{-11}$ to $10^{-10}$ level and pre-rupture nucleation slip in the hypocentral region is constrained to have a moment less than $10^{-5}$% of the mainshock seismic moment [18]. Finally, since these sensors have the capability to record the broad range of earthquake preseismic, coseismic and postseismic signals with submillimetric resolution, they represent a powerful tool for complementing existing networks deployed in active regions.

### 3.2. Strain-Infrasound Coupling: Observation and Theory

Recording of strain waves provides a large range of applications in seismology. In addition to their use in estimating seismic phase velocities [38], modeling earthquake source processes [18,34] or detecting Earth free oscillations [39], dynamic strain recordings have potential for detecting and characterizing atmospheric infrasound generated by earthquakes [40]. Infrasound correspond to the subaudible (<20 Hz) spectrum of acoustic waves and three main types of infrasound are generally observed. Epicentral infrasound is generated near or within the earthquake source region and result from the coupling due to movements of the Earth's surface and energy transfer through the Earth-atmosphere interface [41]. Remote infrasound occurs far away from the seismic source and is caused by seismic waves coupling with infrasound at topography [42] or in sedimentary basins [43]. Finally, near-receiver infrasound represents a ground-coupled acoustic wave that is generated by the vertical ground motion associated with large amplitude seismic waves while passing through an observation site [44].

Figure 9 shows detection of near-receiver infrasound signals by collocated strainmeter and microbarometer stations in the LV. The systematic analyze of atmospheric pressure response to dilatational strain waves during seismoacoustic disturbances associated with Rayleigh waves from $M_w \geq 6$ events reveals that infrasound signals have a phase shift of $-60°$ to $-100°$ with respect to the dilatation strain signals with a coupling factor ranging from 0.002 to 0.006 Pa/n$\epsilon$ [40]. A mean phase value of about $-75°$ with a coupling coefficient of 0.0037 Pa/n$\epsilon$ is inferred among the stations, and coupling parameters show no dependence with the infrasound main periods in the range 1–20 s. While ground-coupled atmospheric infrasound is generated instantaneously by the vertical seismic velocity induced by the passing wave at the station [44], theoretical coupling between atmospheric infrasound (*P*) and dilatational strain ($\epsilon_v$) for a Rayleigh wave propagating near the surface considering an homogeneous elastic half-space implies a phase term, suggesting that infrasound is not generated instantaneously by dynamic strain waves [40]:

$$\frac{P}{\epsilon_v} = \rho c \left( \frac{\gamma}{1 - 2\gamma} \right) \left( \frac{V_r}{0.24} \right) \tan(\omega t - k_x x) \tag{2}$$

where $\rho$ is the density of air ($\rho$ = 1.204 kg·m$^{-3}$ at the temperature of 20 °C), $c$ the velocity of sound in air ($c \simeq 343$ m·s$^{-1}$ at 20 °C), $\gamma$ the Poissons's ratio, $k_x$ the horizontal wavenumber, $\omega$ the wave angular frequency and $V_r$ ($= \omega/k_x$) the Rayleigh wave phase velocity. The strain-infrasound amplitude factor depends not only on the near-surface rock properties (through the Poisson's ratio) but also on the wave period (through $V_r$), as Rayleigh waves are dispersive. The observational mean coupling factor for the LV is in good agreement with the theoretical estimates in the case of typical Poisson's ratio values ($\nu \simeq 0.25$–0.30) and Rayleigh-wave velocities ranging from 3 to 4.5 km·s$^{-1}$ (Figure 10a).

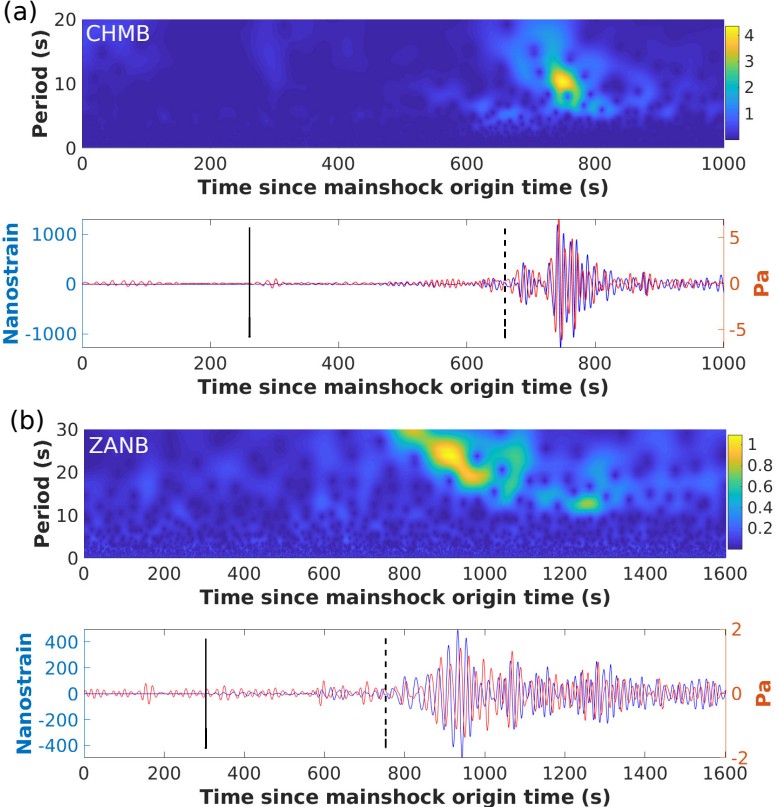

**Figure 9.** Example of detection of infrasound signals during teleseismic events using continuous wavelet transform analysis of the atmospheric pressure time series: (**a**) strain (blue curve) and infrasound (red curve) signals with a main period of 12 s recorded by CHMB during the 12 May 2008 $M_w$ 7.9 Sichuan earthquake and (**b**) signals with a main period of 20 s recorded by ZANB during the 11 March 2011 $M_w$ 9.1 Tohoku earthquake. Strain contraction and atmosphere overpressure are positive. Note that atmospheric overpressure follows contractional strain waves with a slight phase delay. Solid and dashed lines denote the *P*-wave and Rayleigh-wave arrival time at the station, respectively. "A. Canitano, Observation and theory of strain-infrasound coupling during ground-coupled infrasound generated by Rayleigh waves in the Longitudinal Valley (Taiwan), *Bull. Seismol. Soc. Am.*, 2020, 110, 2991–3003, ©Seismological Society of America".

On the other hand, dilatational strain is in phase with the vertical component of the seismic acceleration ($a_z$) at the free surface, with a coupling factor depending on the Rayleigh-wave period (through $V_r$ and the wave pulsation $\omega = 2\pi/T_c$) [40]:

$$\epsilon_v = \left(\frac{0.12T_c}{\pi V_r}\right)\left(\frac{1-2\gamma}{\gamma}\right)a_z \tag{3}$$

Based on a preliminary analysis of strain-acceleration coupling conducted for HGSB station, we found that frequency-dependent coupling parameters show a difference of about one order of magnitude between long-period ($T_c \geq 12$ s) and short-period ($T_c \leq 3$ s) infrasound signals. Observational coupling factors are in good agreement with theoretical estimates for typical Rayleigh-wave velocities and Poisson's ratios for the region (Figure 10b). Therefore, a theoretical phase shift of $-90°$ is expected between strain and vertical seismic velocity signals due to the intrinsic nature of dilatational strain. Overall, preliminary coupling parameters inferred for ground-coupled acoustic waves in the region agree well theoretical estimates. This first attempt to analyze strain-infrasound mechanical coupling shows the potential of strainmeters for seismoacoustic analysis and emphasizes the use of those recordings combined with seismic, geodetic, and infrasonic datasets to help in better understanding the coupling between the Earth's crust and the atmosphere.

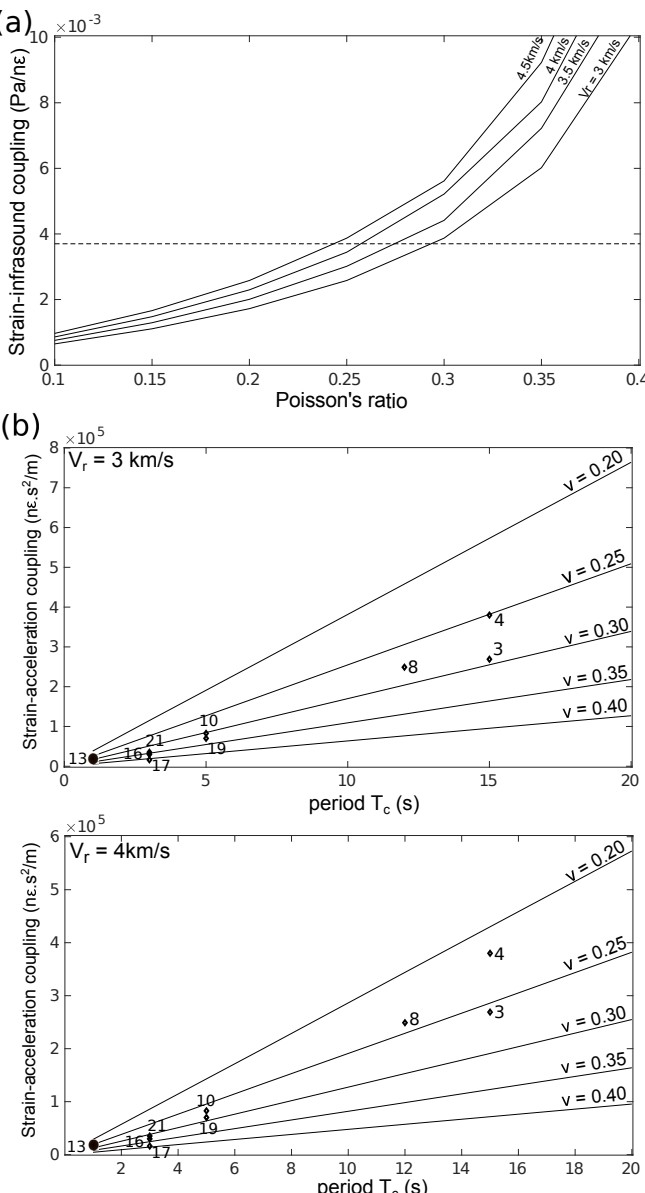

**Figure 10.** (**a**) Theoretical strain-infrasound coupling factor $\rho c \left( \frac{\gamma}{1-2\gamma} \right) \left( \frac{V_r}{0.24} \right)$ as a function of the Poisson's ratio and Rayleigh-wave velocity. The horizontal dashed line depicts the mean coupling factor obtained for the LV (0.0037 Pa/nε). (**b**) Theoretical strain-acceleration coupling coefficient $\left( \frac{0.12 T_c}{\pi V_r} \right) \left( \frac{1-2\gamma}{\gamma} \right)$ as a function of the Rayleigh wave (and infrasound) main period $T_c$, Poisson's ratio and Rayleigh wave velocity. Observational strain-acceleration coupling coefficient (diamonds) are estimated using HGSB strainmeter and a nearby seismometer (vertical seismic acceleration is obtained from the time derivative of the velocity seismogram). "A. Canitano, Observation and theory of strain-infrasound coupling during ground-coupled infrasound generated by Rayleigh waves in the Longitudinal Valley (Taiwan), *Bull. Seismol. Soc. Am.*, 2020, 110, 2991–3003, ©Seismological Society of America".

### 3.3. Evidence for Aseismic Deformation in the LV

Over the last two decades, the growing deployment of seismological and geodetic monitoring arrays in active regions has revealed episodic aseismic slip in the crust spanning timescales ranging from seconds to years [45]. SSEs (originally called slow earthquakes) were first revealed by borehole strainmeters in the Japan Sea [46,47]. Slow events are now widely observed, often accompanied by earthquake swarms or non-volcanic tremors [48,49], and together contribute to the release of the long-term earthquake strain budget in active

regions [50]. Aseismic events are also observed in volcanic regions and are possibly resulting from large rainfall [51].

### 3.3.1. Detection and Characterization of SSEs

Strainmeters offer an ideal complement to traditional geodetic detection techniques, such as GNSS and InSAR, in particular to uncover localized, small-scale and small-magnitude SSEs. They have helped in discovering slow events with limited equivalent geodetic moment magnitude ($M_w \simeq 4.8$ to 5.8) along the San Andreas fault, western USA [52], on the North Anatolian fault, Turkey [53], as well as repeating events in Cascadia modulated by tidal stress [54]. Recently, the deployment of subseafloor borehole strainmeters has also extended the onland sensor detection capabilities to offshore regions [55].

Borehole strainmeter observations have revealed the presence of SSEs along the LVF (Figure 11). Slow events are lasting days to weeks and generate low to moderate strain levels ($\leq$50–100 n$\epsilon$) (Figure 12). SSE sources are located at shallow to moderate depths along the LVF (5 to 15 km), they are associated with geodetic moment magnitude ranging from 4.5 to 5.5 and remain undetected by GNSS stations. Due to their low stress drop nature, SSE occurrence and activity are highly sensitive to small stress variations (a few kPa) [54]. Therefore, static and/or dynamic sources of stress are often shown to trigger SSE activity [55,56] or even arrest an ongoing event [57].

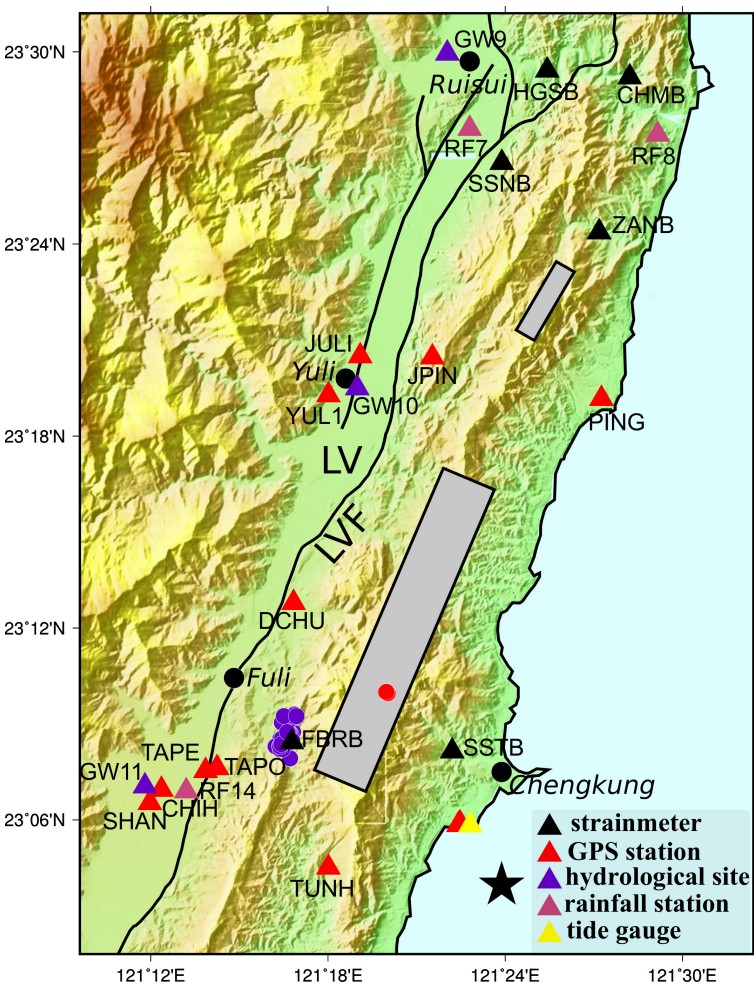

**Figure 11.** Major faults and instrumentation in eastern Taiwan. Large and small gray rectangles denotes the surface projection of the 2011 and 2013 SSE source faults, respectively. Red and blue dots represent the relocated earthquake multiplet and aftershocks that coincide with the 2011 SSE and 2010 afterslip, respectively. Black star denotes the Chengkung earthquake epicenter and black dots the main towns.

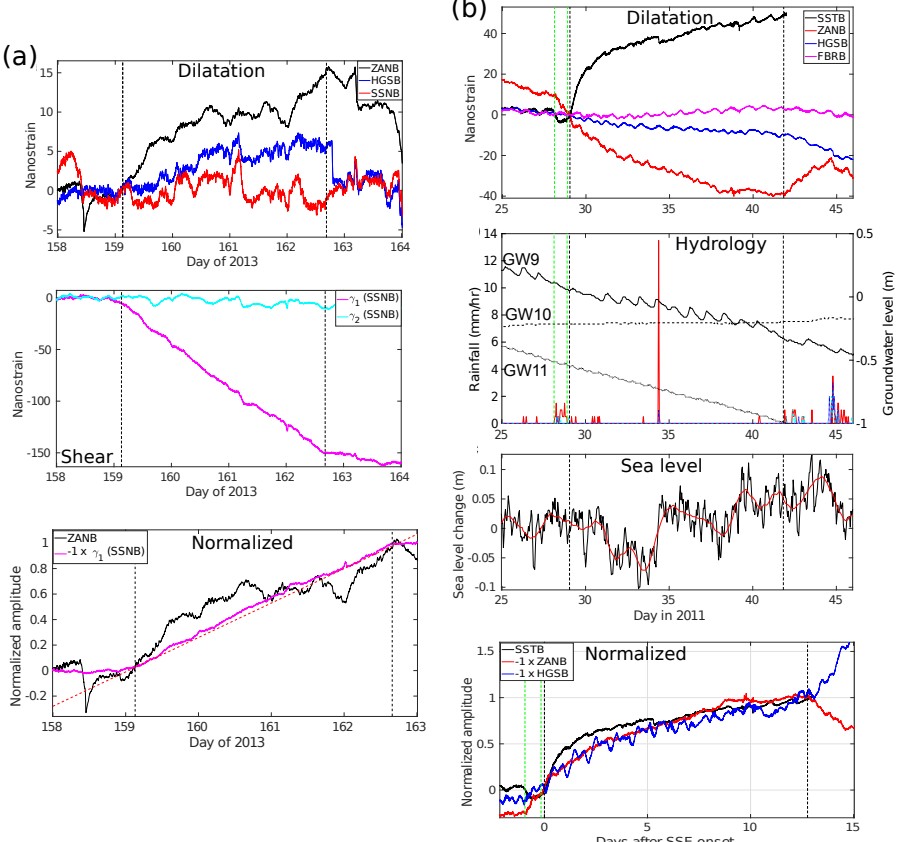

**Figure 12.** Detection of SSEs on the LVF using borehole strainmeters. (**a**) Dilatation and shear signals recorded in the Ruisui area associated with a $M_w \sim 4.5$ SSE in 2013. Figure adapted from Canitano et al. (2019) [5]. (**b**) Dilatation signals, hydrological perturbations (groundwater level: black curves, cumulative rainfall: red, blue and magenta bars) and sea level changes during the 2011 $M_w$ 5.5 SSE. Normalized strain time-series are signals normalized by the value reached at the end of the SSE episode. Vertical black and green dashed lines indicate the duration of SSE and light rainfall episodes, respectively. Figure adapted from Canitano et al. (2021) [6].

SSEs in the LV have static stress drops ranging from a few kPa to a few hundreds of kPa. The June 2013 SSE initiated during a phase of maximum tidal Coulomb stress changes on the LVF ($\sim$1–1.5 kPa) (Figure 13). SSEs can be triggered by various exogenous factors among which tidal Coulomb stress changes only account for a few percent of the SSE stress buildup [5]. Therefore, the influence of tidal stress on SSE generation in central LV yet remain to be analyzed carefully. The 2011 $M_w$ 5.5 SSE occurred in the near-source region of the 2003 $M_w$ 6.8 Chengkung earthquake (Figure 11), about 7 years following the mainshock. The slow event may have triggered a burst of repeating earthquakes and its occurrence is strongly related to the significant stress changes imparted by the combination of coseismic and postseismic slip following the mainshock ($\sim$0.5–1 MPa) [6]. Coseismic and postseismic perturbations induce permanent stress changes, with effects lasting from years to decades, and can thus trigger seismic or aseismic deformation long after the mainshock occurrence [58]. It is worth noting that the SSE source is localized in a region where about 0.2 to 0.4 m of slip deficit was accumulated from 1997 to 2011 mainly due to insignificant coseismic slip ($<$0.05 m) and moderate postseismic slip ($\sim$0.2–0.3) (Figure 14). A limited fraction (about 15–35%) of the slip deficit has been accommodated by the SSE while additional aseismic transient slip should likely help to release the remaining slip deficit.

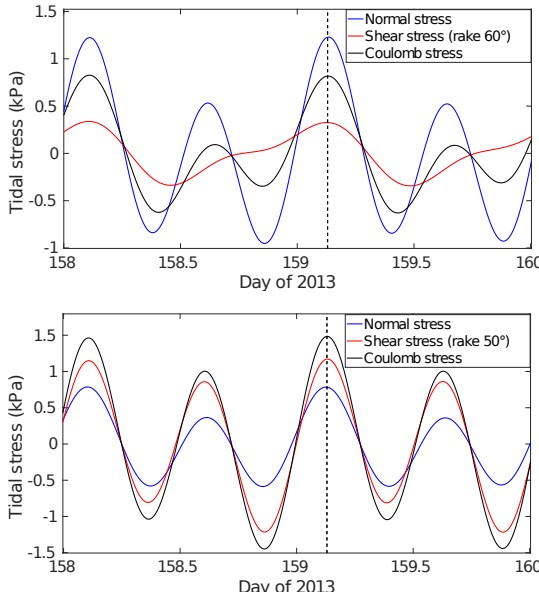

**Figure 13.** Modeling of solid Earth and ocean tidal stress perturbations on the fault plane candidates of the June 2013 SSE. Coulomb stress changes in the direction of geological slip on the plane (rake = 60–70°) are maximal at the time of the SSE initiation (depicted by vertical black dashed line). Figure adapted from Canitano et al. (2019) [5].

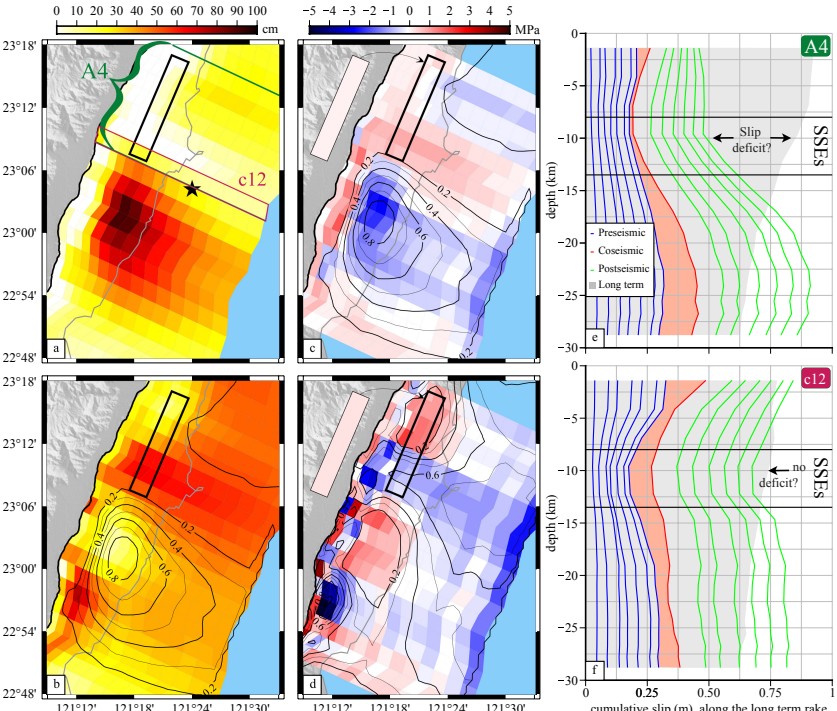

**Figure 14.** Kinematic slip models of (**a**) the 2003 Chengkung earthquake and (**b**) the following 7-year postseismic period and static Coulomb stress changes imparted by (**c**) coseismic and (**d**) postseismic slip, resolved onto the LVF fault plane. The plane depicted on the side represents Coulomb stress changes computed for a fixed rake of 70°. The black rectangle outlines the SSE rupture area and black star denotes the Chengkung earthquake epicenter, respectively. Black curves indicate contour lines of the coseismic (**b**,**c**) and postseismic (**d**) slip distribution models (in meter). Cumulative slip along the long-term rake for area A4 and c12 column (see (**a**)) are given respectively in (**e**,**f**). Red and gray shading give respectively the coseismic slip and the total motion of the fault if it had crept at the long-term slip rate from 1997 to 2011. Figure adapted from Canitano et al. (2021) [6].

### 3.3.2. Afterslip Detected during Small Earthquakes ($M_w < 5$)

Afterslip is a phase of earthquake postseismic relaxation and occurs on the mainshock fault plane or surrounding faults in response to stress changes resulting from the coseismic stress drop [59]. Postseismic slip following moderate to large shocks (typically $M_w \geq 6$) has long been detected by GNSS or InSAR in a variety of tectonic regions [60,61] and represents a significant fraction of the total energy budget of earthquake sequences [62]. In the case of smaller events ($M_w < 6$), recent studies show that aseismic moment is comparable to, or even larger, than the seismic moment [63,64] and the smaller the earthquake is, the larger is its aseismic contribution [63]. Such a behavior is consistent with the predictions from rate-and-state friction laws, since the ratio of aseismic to seismic moment is expected to increase if the size of the asperity decreases. Therefore, the analysis of postseismic phases for small earthquakes is important for assessing earthquake hazard but their detection remains challenging.

Figure 15a depicts the volumetric strain variations recorded by FBRB station during a 1-month long afterslip following a $M_L$ 5 earthquake in the Fuli region in April 2010 (Figure 11). This signal represents the largest and longest strainmeter detection of afterslip associated with a small earthquake sequence in the LV to date [19]. Additional features of aseismic slip are revealed by the presence of repeating seismicity near and within the source region and by the control of aftershock productivity through brittle creep. Afterslip-driven aftershocks are often detected during large earthquakes [60,61], but this is the first evidence for seismic-aseismic interplay in the case of small-magnitude earthquakes ($M_w \leq 5$). Besides, the geodetic moment of the postseismic phase (4.8 $\pm$ 0.2) is nearly equivalent to the coseismic moment of the sequence, suggesting a large contribution of aseismic slip during small-magnitude seismic sequences.

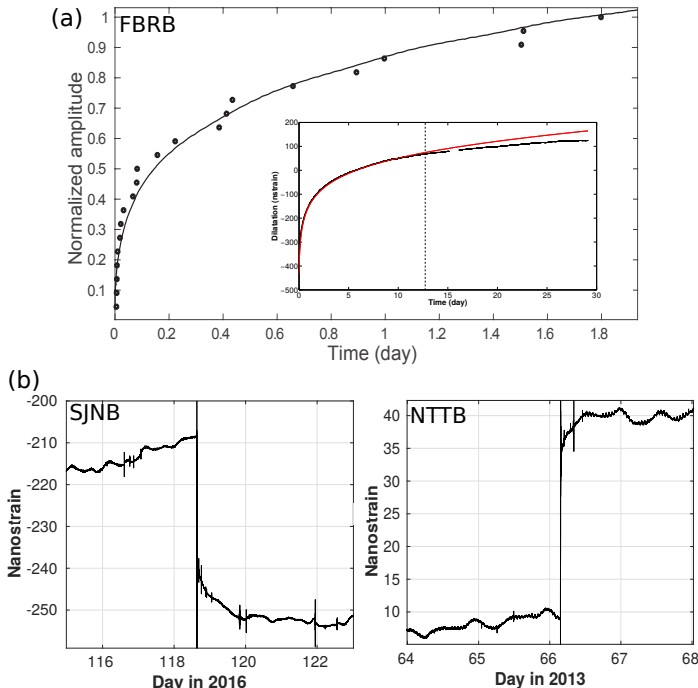

**Figure 15.** (**a**) Evolution of the cumulative number of aftershocks (black dots) with dilatation signal recorded by FBRB during the early stage of postseismic deformation (first 2 days). Signals are normalized by their maximal amplitude after 2 days. (Inset) Afterslip signal (black curve) and fit with a rate-dependent frictional model from (red curve) (Equation (4)) over a 1-month period. The dashed vertical line denotes the limit (first 13 days) for a satisfactory fit between observation and model that covers about 85–90% of the aseismic slip evolution. Figure adapted from Canitano et al. (2018) [29]. (**b**) Observations of short-lived (1–2 days) afterslip: (left) $M_L$ 5.7 earthquake at 26 km from SJNB and (right) $M_L$ 5.9 earthquake at 17 km from NTTB. Expansion is positive.

Rock frictional parameters and regional stress estimates can be inferred by modeling the afterslip signal using a rate-dependent friction law [60]. The afterslip evolution $\epsilon_a$ is estimated by adjusting the geodetic postseismic frictional law [65] to volumetric strain time-series [19] (Figure 15a (inset)):

$$\epsilon_a(t) = \epsilon_0 t_r \log(1 + d(\exp(t/t_r) - 1)) \tag{4}$$

where $\epsilon_0$ denotes the volumetric long-term strain rate, $t_r = A\sigma_n/\dot{\tau}$ is the relaxation time and $d = \exp(\Delta\sigma/A\sigma_n)$ ($\sigma_n$ is the normal stress, $A$ is a rheological parameter and $\dot{\tau}$ the interseismic shear stress) is the velocity jump due to the coseismic shear stress change $\Delta\sigma$ on the fault [60]. Parameter estimates ($\epsilon_0 = 2 \times 10^3$ n$\epsilon$·yr$^{-1}$, $A = 3 \times 10^{-4}$, $\dot{\tau} \simeq 0.3$ MPa·yr$^{-1}$) were in good agreement with previous studies in the region [65]. Finally, additional observations are shown in Figure 15b illustrating the high-resolution strainmeter detection capabilities to record continuously abrupt changes (coseismic steps) and subsequent slow variations.

### 3.4. Ground Deformation Detected during Tropical Cyclones

Although borehole strainmeters have been extensively used for observing tectonic processes, they are sensitive enough to monitor surface loads and other non-tectonic deformation. A good example is the clear deformation that is recorded when tropical cyclones (also named typhoons in Asia) make landfall in Taiwan, about 6 times per year on average [66]. The strainmeter network in Taiwan offers a unique opportunity to observe such events. Indeed, the spatial density of the instruments allows redundant observations of the same typhoons yet not exactly in the same conditions (not the same watershed). This guarantees the robustness of the hypothesis tested in the modeling of the deformation. Moreover, the instruments are located on the eastern coast of Taiwan where they can start sensing the typhoon earlier and stronger than anywhere else in Taiwan, since most typhoons come from the East and thus make landfall in eastern Taiwan (Figure 16a), before loosing intensity. Such extreme meteorological events are characterized by a significant air pressure drop and, most of the time, intense rainfalls. The air pressure drop creates ground expansion that reaches about 50 to 100 n$\epsilon$. The peak of expansion is in fact not often reached since concomitant rainfalls start to build up a significant load on the crust surface. That rainwater load compresses the ground, hence counter-balancing the expansion due to the air pressure drop. In fact, contraction exerted by the rainwater load often accounts for most of the deformation, reaching several hundreds of n$\epsilon$ (Figure 16b). This contraction is slowly relieved after the typhoon has passed, yet other new processes and rainfalls may prevent from observing a clear return to the state of contraction prior to the typhoon.

The ground expansion is synchronous with the air pressure variation but the rainwater loading keeps building up even after the rain has stopped. Figure 17 shows that modeling the strain only from the precipitation accumulated above the strainmeter sensor fails to explain both the amplitude and the dynamics of the observed deformation. Indeed, the computed compression starts a few hours to one day before the observed one and it underestimates it by 15 to 20%. Indeed, the rainwater is drained over the entire watershed and needs time to flow from the precipitation area down to the valleys, where the strainmeters are located. This process also allows to accumulate more rainwater above the strainmeter than the one that precipitated locally. As a result, the time delay between the rainfalls and the compression reflects both the size of the watershed (rainwater takes more time to run off from large watershed) and the ability of the rainwater to flow on the mountain slopes [22]. The strainmeter data thus allows to quantify the amount of rainwater that loads the ground and its dynamics.

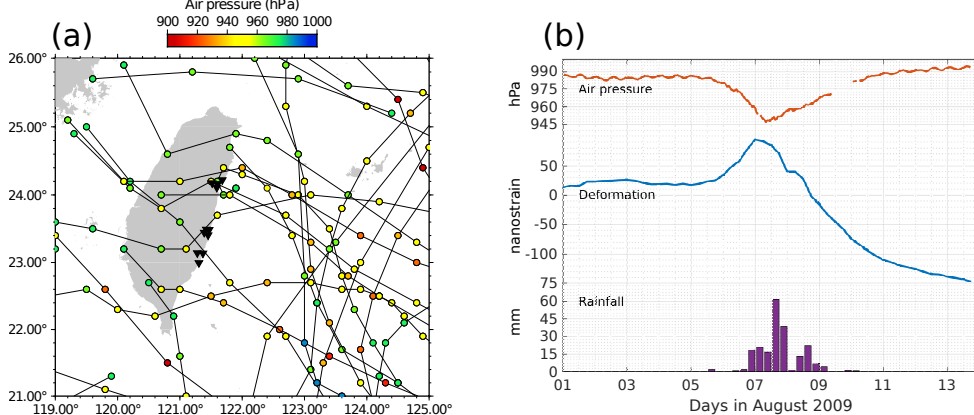

**Figure 16.** (**a**) Trajectories of typhoons and super typhoons in the Taiwan area between 2010 and 2019 [67]. The dots are colored according to the surface air pressure, reported every six hours. These tropical cyclones initiate in the east and move westward, either crossing Taiwan or running along its eastern coast. The strainmeter network (inverted black triangles) can monitor their effect on the crust. (**b**) Typical dilatation record (blue curve, middle) when a typhoon makes landfall, in this case typhoon Morakot (August 2009) observed at ZANB (signal is calibrated and detided). Note that we removed linear long-term trends that may run across several weeks before and after the typhoon, to focus on the typhoon process alone. The ground first expands (deformation increases) due to the air pressure drop (red curve, top) then contracts due to the precipitations (purple bar plot, bottom) that create a rainwater surface load.

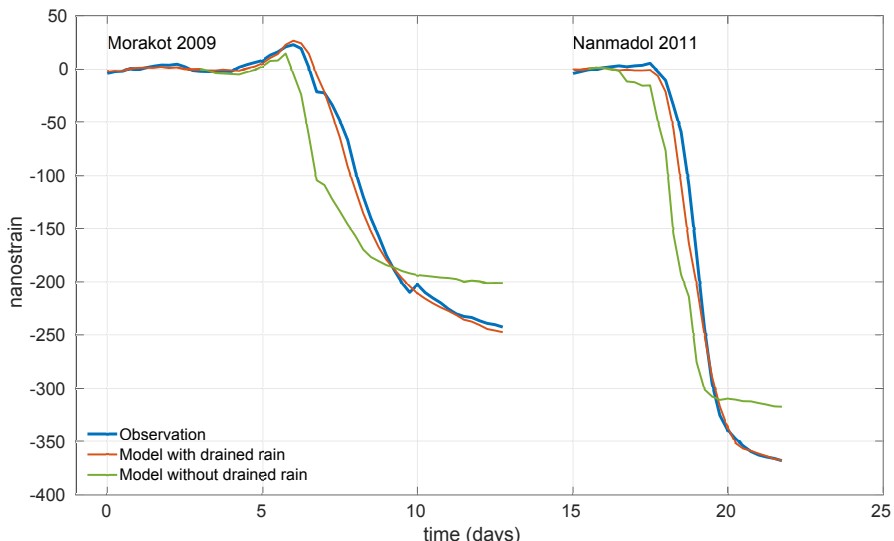

**Figure 17.** Strain variations measured during typhoons Morakot in 2009 and Nanmadol in 2011, at site TRKB (blue curve). This figure shows the necessity to account for the load created by rainwater drained from watershed in addition to the rainwater falling directly nearby the strainmeter. The strain computed in that case (red curve) explains the strain observations better than the strain computed only from the accumulated precipitation on the ground above the sensor (green curve).

## 4. Future Perspectives

### 4.1. Hydrological Modulation of Crustal Strain, Seismicity and Aseismic Slip

Long-term strain accumulation, fault slip rates and seismicity in active regions are mainly driven by plate tectonics. However, several surface or subsurface processes (e.g., changes in water storage, rainfall, air pressure variations) can alter the stress in the crust [68], and thus affect the long-term tectonic deformation, modulate seismic activity or even trigger earthquakes [69,70]. Redistribution of surface loads takes place over a broad range of spatial scales (from a few km to thousands of km) and produces geodetically

observable deformation [68,71]. In Taiwan, the hydrosphere exhibits a strong annual cycle
following the dry (October to April) and wet (May to September) season precipitation.
Continental water and groundwater storage increase during the wet season, enhanced
by the heavy rainfall brought by tropical typhoons and monsoons. The seasonal water
cycle generates annual variations of 5 to 15 m in groundwater levels, inducing elastic
deformation of the Earth crust of 5 to 20 mm in GNSS vertical displacement time-series [72]
(Figure 18a). Given the mean annual water-thickness change of about 0.7 m in eastern
Taiwan [72], the estimated seasonal volumetric strain change is about 100–150 n$\epsilon$ in eastern
Taiwan (Figure 18b), less than the observed annual volumetric strain changes of 0.3–1 $\mu\epsilon$
at strainmeter sites (Figures 18a and 19). The model predicted annual strain change is
underestimated and is likely due to a smooth soliton of seasonal water storage changes
inferred by GNSS.

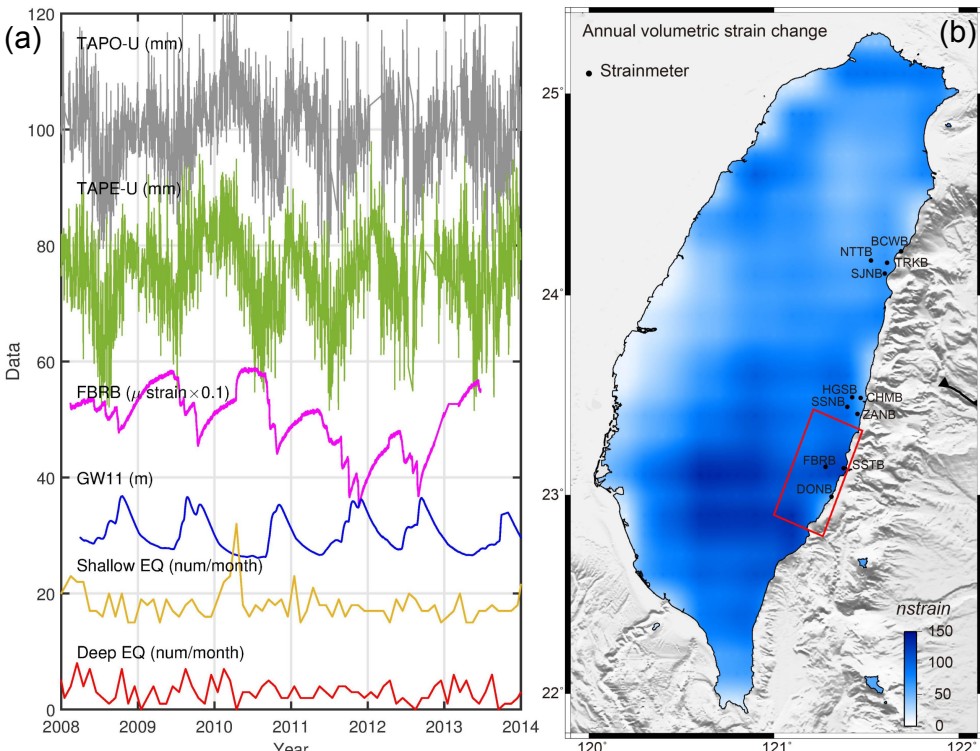

**Figure 18.** Annual variations due to hydrological cycles. (**a**) GNSS vertical position time series at
sites TAPE and TAPO across the LVF (see Figure 2 for station locations). Magenta and blue lines show
the time series of volumetric strain at FBRB and groundwater level at the station GW11. Orange
and red lines indicate shallow and deep earthquakes in the southern LVF (red box in (**b**)). The
depth threshold is about 17 km which divides the declustered catalog using the nearest neighbor
algorithm [73] into similarly sized numbers of events. (**b**) Volumetric strain change at the depth
of 1 km due to seasonal hydrological loading. Change are estimated using a 1D spherical layered
Earth model [74] and multiple vertical source loading [70,75] with amplitudes constrained by disk
loads [72].

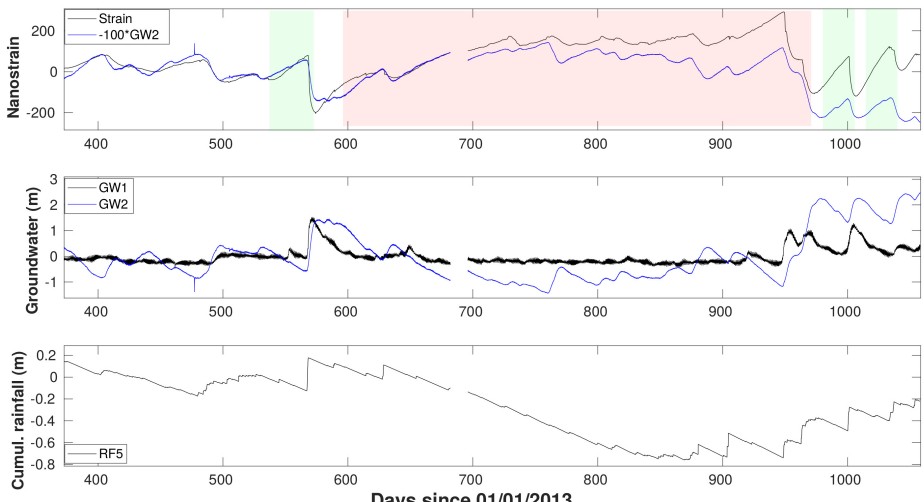

**Figure 19.** Dilatational strain (expansion > 0), groundwater level changes and detrended cumulative rainfall recorded in the Hualien-Taroko region by station SJNB (see Figure 2 for station locations). From top to bottom: dilatation (black curve) and groundwater level (station GW2, signal is inverted and multiplied by an arbitrary factor for comparison); groundwater level and detrended cumulative rainfall. Red and green shaded boxes show crustal strain modulation by groundwater level variations at an annual and bimonthly period, respectively.

At the seismogenic depth of 5–40 km in eastern Taiwan, estimated amplitude of annual Coulomb stress is about 2–5 kPa on N-S trended, 30° east-dipping receiver fault geometry with a rake of 90°. Hsu et al. (2021) [73] demonstrate clear evidence for synchronized and asynchronous earthquakes with water unloading cycles in eastern Taiwan. Annual seismicity rates of deep earthquakes are primary driven by elastic hydrological load cycle, whereas the shallow seismicity rates are likely related to fluid transport, pore pressure changes, and other time-dependent processes [73]. Despite a small annual hydrological stress change with amplitudes slightly larger than the tidal stress change in Taiwan (∼1–1.5 kPa [5]), hydrological loading can modulate regular earthquakes whereas no seismicity-rate variations are observed at tidal periods. Laboratory experiments show earthquake triggering is expected to be most significant at particular time periods comparable to the duration of earthquake nucleation [76]. Seismicity also exhibits period-dependent response to periodic stress perturbations [77] which shows that the period of greatest correlation for natural faults is near 1 year and explains the modulation of seismicity by annual hydrological loading. Therefore, an unambiguous determination of the possible relationship between seismicity and hydrological loading cycles over multiple spatiotemporal scales and other natural processes with various forces and stressing periods can provide valuable insights into earthquake triggering mechanisms. Comparisons of spatial and temporal patterns for various observations including borehole strainmeter records, groundwater level, GNSS time-series, *b*-values in the Gutenberg-Richter law, shear wave velocity changes derived from seismic ambient noises are crucial in evaluating the parameter sensitivity to each individual physical process. Understanding earthquake triggering mechanisms is paramount to reducing seismic hazards.

Besides, a better knowledge of the dominant source or combination of sources of hydrological deformation is also important for improving our understanding of crustal dynamics and to improve the removal of seasonal effects for studies focusing on tectonic and other non-seasonal processes. This would be fundamental to detect and constrain aseismic sources of deformation, the mechanisms under which aseismic slip occur and evolve, and its contribution to the earthquake cycle of the LVF.

### 4.2. Quantification of the Mass of Sediment Removed by Landslides

The Taiwan orogen experiences numerous landslides triggered by earthquakes [78] and heavy rainfalls, often during typhoons [79]. They have catastrophic societal consequences, can significantly alter the landscape and it has been recently shown that they also influence the seismicity [80]. The latter is suggested to be due to the mass removal from the land, which unclamp the faults beneath [81]. Thus, quantifying the mass of sediment removed by landslides is important to monitor and assess erosion processes as well as seismicity variations. Strainmeter measurements, thanks to their sensitivity to surface loads, can therefore be very useful in such a quantification. One drawback is that they are rather sensitive to surface loads located in their vicinity, and the ideal case is when the surface load is just above them. But since the network is spread on several watershed in areas prone to landslides, such effect should be observable, with an order of magnitude from tens to hundreds of n$\epsilon$. In fact, the main challenge would be to separate the sediment load from other effects able to deform the ground, and the sections above show that they are numerous. A strategy would be to combine the strain measurements with surface measurements (e.g., photogrammetry) in order to better constrain the location of the sediment load. Since the eroded sediment are gradually flushed to the sea, a decay of the surface load effect should also appear in the time-series. This effect could be first investigated after massive landslides, which sediment deposits should create significant deformation lasting over several years.

### 4.3. Rapid Moment Magnitude Estimate and Implications for EEW Systems

EEW systems utilize the difference in propagation between *P*- and *S*-waves to forecast areas at risk of potential destruction by earthquakes. They aim to provide a time-window to act before strong shaking happens to help alleviate damages and injuries. Well-functioning systems strongly rely on fast and robust estimates of earthquake magnitudes, source locations and ground motions [82]. As a major component of EEW networks, both seismometers and high-rate GNSS sensors may however show limitations in recording near-source large ground motions with high-resolution [83,84]. Strainmeters have the capability to record the whole spectrum of elastodynamic deformation (static and dynamic) [34] with little noise level, without saturation, and with no additional signal processing, and therefore show potential for EEW applications [82]. An earthquake magnitude scale based on local broadband dynamic strain waveforms has recently been developed by Barbour et al. (2021) [85]. Its application to the 2019 $M_w$ 6.4 and 7.1 Ridgecrest earthquakes showed very promising results with magnitude estimates being in agreement to within about 0.3 magnitude units [85]. Therefore, the strainmeter networks may have a role to play in future EEW systems in Taiwan [86], provided a robust magnitude-scale calibration can be derived from past near-source large events (e.g., 2013 Ruisui, 2018 Hualien).

### 4.4. Tidal Variations

All the studies discussed so far only account on tides in order to compute the calibration factor of each strainmeter. After that, the tidal signal is removed from the observations in order to emphasize deformation due to other processes. However, the tidal signal itself can be worth studying. This has been done using long time-series of superconducting gravimeters installed in Europe, which mainly shown the seasonal variations of tidal parameters [87,88]. Superconducting gravimeters must also be calibrated, and that calibration is done using tides, just like for strainmeters. Meurers et al. (2016) [87] review five reasons to explain such variations, three of which relate to non-geophysical process (calibration, data pre-processing and numerical artifacts), one to temporal variation of the ocean load and one to tectonic processes [89], which influence gravity as well as displacement and strain. Therefore, given the vigorous tectonic activity at work in and around the Taiwan area, it would be interesting to investigate such tidal variations in a network of strainmeters.

## 5. Conclusions

In this paper, we showed that the Taiwanese network of *Sacks–Evertson* borehole strainmeters significantly contributed to the analysis and understanding of a wealth of geophysical processes. These processes are mostly related to the sustained tectonic activity at work in Taiwan. They provide information about earthquakes, ranging from their source mechanism to their atmospheric infrasound signature. In addition, the high sensitivity of strainmeters at short- to intermediate-period (days to months) make them significantly valuable for monitoring more slow and quiet processes, such as SSE, creep and afterslip with limited magnitude (typically $M_w \leq 6.0$) that are unresolved by surface geodesy. Besides, non-tectonic deformation processes due to hydrology and typhoons also build up unambiguous signals in the strainmeter data, which allow us to better quantify the influence of surface loads on crustal deformation. Despite the efforts required for their installation and calibration, we believe that this study demonstrates how such instruments can benefit research in a wide range of geophysical processes and the interactions between them.

**Author Contributions:** Conceptualization, A.C., M.M. and Y.-J.H.; resources, A.L., S.S. and H.-M.L.; writing—original draft preparation, A.C., M.M. and Y.-J.H.; writing—review and editing, A.L. and S.S. All authors have read and agreed to the published version of the manuscript.

**Funding:** This research was funded by Ministry of Science and Technology grant MOST 108-2116-M-001-027-MY2.

**Institutional Review Board Statement:** Not applicable.

**Informed Consent Statement:** Not applicable.

**Data Availability Statement:** The data presented in this study are available on request from the corresponding author.

**Acknowledgments:** We are thankful to the Assistant Editor David Hu and the three anonymous reviewers for their constructive suggestions and comments allowing to improve the manuscript. We thank the support staff of the Carnegie Institution of Washington for the construction, installation, and maintenance of the dilatometers. We are grateful to our colleagues at the Institute of Earth Sciences, Academia Sinica who have participated in collecting strainmeter data. This is the contribution of the Institute of Earth Sciences, Academia Sinica, IESAS2397.

**Conflicts of Interest:** The authors declare no conflict of interest.

## Abbreviations

The following abbreviations are used in this manuscript:

| | |
|---|---|
| GNSS | Global Navigation Satellite System |
| LV | Longitudinal Valley |
| LVF | Longitudinal Valley Fault |
| SSE | Slow Slip Event |
| InSAR | Interferometric Synthetic Aperture Radar |
| EEW | Earthquake early warning |

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
