# Peer review of "Fifteen Years of Continuous High-Resolution Borehole Strainmeter Measurements in Eastern Taiwan: An Overview and Perspectives"

_2624-795X, doi:10.3390/geohazards2030010_

Round 1

Reviewer 1 Report

The goal of the present manuscript is to present the “Fifteen Years Of Continuous High-Resolution Borehole Strainmeter Measurements In Eastern Taiwan: An Overview And Perspectives”

Overall, I think this is a really interesting paper with enough data and sufficient and informative figures. The manuscript is well written and easy to follow. In general, I am satisfied with the structure of the paper even tough that it does not follow the common layout (Introduction, Study area, Methodology, Results, Discussion, Conclusions). It would be better though if you could add a small paragraph as Conclusions. In my opinion this is important since this is the main idea where a reader will understand.

Furthermore, I would like to ask if you have detected any anomalies to borehole strainmeters before an earthquake event. For example, Yu et al. (2021) tried to extract pre-earthquake anomalies based on a borehole strain network. Have you identified something similar, apart from the short-term prediction you mention in 4.3? Perhaps you could add a few words, maybe as subchapter 4.5

Author Response

Dear Reviewer,

we appreciate your comments, thank you very much.

  • We added a Conclusion section to the manuscript.
  • Yes, we analyzed pre-earthquake signals during the 2013 Ruisui earthquake, but did not find any changes. We added a few details about that at the end of Section 3.1 (P. 9, L. 194-198).

Reviewer 2 Report

In this paper, the authors present an overview of the main advances enabled by continuous strain measurements in eastern Taiwan, and propose a set of future research directions that address recent challenges in seismology, hydrology and crustal strain modeling.

The introduction provides sufficient background but I suggest to include a few references in order to have a more complete picture of the state of art.

The authors have well schematized the work: it is well integrated in the general context of the strainmeter scientific area and offers useful suggestions for future analyses. I find the manuscript suitable for publication after a few minor revisions.

Comments

-Introduction

page 2, line 41: I suggest to include the references Giudicepietro et al 2020 and Di lieto et al. 2020.

Giudicepietro, F., López, C., Macedonio, G., Alparone, S., Bianco, F., Calvari, S., et al. (2020). Geophysical precursors of the July–August 2019 paroxysmal eruptive phase and their implications for Stromboli volcano (Italy) monitoring. Scientific Reports, 10, 10296. https://doi.org/10.1038/s41598‐020‐67220‐1.

Di Lieto, B., Romano, P., Scarpa, R., Linde, A. T. (2020). Strain signals before and during paroxysmal activity at Stromboli volcano, Italy. Geophys. Res. Lett., 47, e2020GL088521, https://doi.org/10.1029/2020GL088521.

-3.4. Ground deformation detected during tropical cyclones

page 17, line 364-365: the word 'indeed' is repeated. I suggest to change the text.

Note for the figures

I suggest to improve the style of the figures, for instance, uniforming the font size of the text and the usage of the parenthesis to indicate the subplots (in fig 16 the a and b subplots are indicated differently respect with the other figures)

Author Response

Dear Reviewer,

we appreciate your comments, thank you very much.

  • We added the proposed references in the Introduction section.
  • We corrected the sentence in P. 17.
  • We modified Fig. 16 for consistency with other figures.

Reviewer 3 Report

Main Comments

Introduction

  • The list of Sacks-Evertson borehole strainmeter installations is missing those deployed in Italy (Campi Flegrei, Vesuvius, Stromboli and Etna). Specifically, I suggest to consider the following manuscripts as a further reading, to show how dilatometers can better constrain deformation sources:
    • Amoruso, A., Crescentini, L., Linde, A. T., Sacks, I. S., Scarpa, R., and Romano, P.: A Horizontal Crack in a Lay- ered Structure Satisfies Deformation for the 2004-2006 Up- lift of Campi Flegrei, Geophys. Res. Lett., 34, L22313, https://doi.org/10.1029/2007GL031644, 2007.
    • Amoruso, A., Crescentini, L., Scarpa, R., Bilham, R., Linde, A. T., and Sacks, I. S.: Abrupt magma chamber contraction and micro- seismicity at Campi Flegrei, Italy: Cause and effect determined from strainmeters and tiltmeters, J. Geophys. Res.-Sol. Ea., 120, 5467–5478, https://doi.org/10.1002/2015JB012085, 2015.

Borehole strainmeter observations in eastern Taiwan

  • In introduction of paragraph “Evidence for aseismic deformation in the LV” it could be mentioned the case of volcanic regions where aseismic transients mostly due to rainfalls have been evidenced by strainmeters. Specifically, I suggest to consider the following manuscript:
    • Di Lieto, B., Romano, P., Bilham, R., and Scarpa, R.: Aseismic strain episodes at Campi Flegrei Caldera, Italy, Adv. Geosci., 52, 119–129, https://doi.org/10.5194/adgeo-52-119-2021, 2021.
  • On page 14, lines 280 and following: probably the sentence should be rephrased and clarified. It seems like that, despite tidal Coulomb stress can trigger SSEs, it will not do. Maybe it could be specified that SSEs can be triggered by various exogenous factors, among which tidal Coulomb stress accounts for a few percent, also if this aspect has yet to be analyzed.
  • On page 17, Figure 16(b) depicts Deformation vs Air Pressure vs Rainfall: is the Deformation related with the measurement of dilatation or shear strain? Which sensor data came from? Are those data raw or somehow filtered?
  • On page 18, Figure 17 shows a significant contribution of strainmeter data to typhoons passing thereby, however it is not clarified neither in the text nor in the Figure caption if the shown offset is canceled once rains flow to watershed, or not. Could you please specify?

Minor Comments

  • Page 6 – Line 133: substitute “that is, mainly” with “that is mainly”
  • Page 6 – Line 134: substitute “dilatational strain, and vice versa” with “dilatational strain and vice versa”
  • Page 9 – Line 189: substitute “S-waves; their” with “S-waves, their”
  • Page 13 – Figure 12 caption: substitute “[5]. ((b) Dilatation” with “[5]. (b) Dilatation”
  • Page 19 – Line 400: substitute “load cycle; whereas” with “load cycle, whereas”

  • In the References section, the referenced article #24 is never mentioned in the main text.

Author Response

Dear Reviewer,

we appreciate your comments, thank you very much.

  • We added the list of strainmeter deployments in Italy as suggested (P2., L. 40-42).
  • We added the reference in Sec. 3.3 (P.12, L. 265-266).
  • We rephrased the sentence on P. 14, L. 280 (P. 14, L.288-291).
  • P. 17, Fig. 16: the deformation is related to dilatation recorded by ZANB. . No filter but we removed linear long-term trends that may run across several weeks before and after the typhoon, to focus on the typhoon process alone. We added these details on Fig. 16(b) caption.
  • P. 18, Fig. 17:  That is right. Indeed the offset will be gradually removed after the typhoon, as the rainwater is evacuated toward the sea or infiltrates in deeper aquifers. We clarified this (P17, L. 362-364).
  • We corrected the minor comments and added the link to reference [28] (previously [24]).